# Human–Animal Interactions in Dairy Goats

**DOI:** 10.3390/ani13122030

**Published:** 2023-06-19

**Authors:** Francesca Carnovale, Giovanni Marcone, Francesco Serrapica, Claudia Lambiase, Emilio Sabia, David Arney, Giuseppe De Rosa

**Affiliations:** 1Chair of Animal Nutrition, Institute of Veterinary Medicine and Animal Sciences, Estonian University of Life Sciences, Kreutzwaldi 1, 51014 Tartu, Estonia; giovanni.marcone@emu.ee (G.M.); david.arney@emu.ee (D.A.); 2Dipartimento di Agraria, Università degli Studi di Napoli Federico II, Via Università 133, 80055 Portici, Italy; francesco.serrapica@unina.it (F.S.); claudia.lambiase@unina.it (C.L.); giuseppe.derosa@unina.it (G.D.R.); 3Scuola di Scienze Agrarie, Forestali ed Ambientali, Università degli Studi della Basilicata, Via dell’Ateneo Lucano 10, 85100 Potenza, Italy; emilio.sabia@unibas.it

**Keywords:** dairy goats, human–animal relationship, avoidance distance, behaviour

## Abstract

**Simple Summary:**

Human–animal interactions may have a strong impact on animals’ living conditions and are fundamental to improving farm animal welfare. Fourteen dairy goat farms located in the province of Potenza (Southern Italy) with flocks ranging in size from 67 to 450 lactating goats were used. The experimental protocol included three tests and observations in the milking parlour: avoidance distance in the pen, avoidance distance at the manger, approach test within 2 min and behavioural observations of stockmen and animals during milking. The percentage of goats allowed to be touched at the manger was higher than that found in the pen, and the avoidance distance at the manger was shorter than that recorded in the pen. The percentage of neutral stockman interactions was positively correlated with the percentage of goats moving when approached at a distance >1 m and was negatively correlated with the animals touched at the manger, in the pen and within 2 min. In conclusion, avoidance distance at the manger and in the pen are valid assessments for evaluating the human–animal connection in goats. Furthermore, avoidance at the manger may reliably replace the pen test.

**Abstract:**

It is widely assumed that the quality of human–animal interactions may have a strong impact on animals’ living conditions and is fundamental to improving farm animal welfare. This work aims to evaluate the effectiveness of methods for assessing and monitoring the welfare of lactating goats. In particular, attention was paid to the methods regarding the assessment of the human–animal relationship. The experimental protocol included three tests and observations in the milking parlour, namely: avoidance distance in the pen, avoidance distance at the manger, approach test within 2 min and behavioural observations of stockmen and animals during milking. Fourteen dairy goat farms located in the province of Potenza (Southern Italy) with flocks ranging in size from 67 to 450 lactating goats were used. All farms raised the Rossa Mediterranean goat breed. The percentage of goats that permitted themselves to be touched at the manger (mean ± SD: 12.36 ± 9.50) was higher than that found in the pen (9.67 ± 11.86) and within 2 min inside the pen (8.19 ± 13.78). The avoidance distance at the manger (0.63 ± 0.28 m) was shorter than that recorded in the pen (1.19 ± 0.58 m). Avoidance distance at the manger was positively correlated with that in the pen (Spearman correlation test (*r_s_*) = 0.607; *p* < 0.01), as was the percentage of goats touched at the manger and in the pen (*r_s_* = 0.647; *p* < 0.01). Approximately 60% of the stockman interactions observed during milking were neutral, while positive and negative interactions had similar values to each other, approximately 20%. The number of neutral stockman interactions was positively correlated with the percentage of goats moving when approached at a distance >1 m (*r_s_* = 0.799; *p <* 0.001) and was negatively correlated with the animals touched at the manger (*r_s_* = −0.607; *p* < 0.05), in the pen (*r_s_* = −0.613; *p* < 0.05) and within 2 min (*r_s_* = −0.669; *p* < 0.01). As regards the degree of association between the behaviour of the milker and the animals during the milking routine, the percentage of neutral interactions tended to be positively correlated only with the number of kicks performed by the animals (*r_s_* = 0.476; *p* < 0.10). It is concluded that avoidance distance at the manger and in the pen, as for other farm animals, are valid tests to evaluate the human–animal relationship in goats. In addition, avoidance distance at the manger may reliably replace the test performed in the pen.

## 1. Introduction

Even after centuries of domestication initiated about 10,000 years ago [1], it can be assumed that prey species, such as goats, may nevertheless perceive humans as predators to be feared, although close and frequent contact may change this perception [2].

Human–animal relationships can be defined as “the degree of relation or distance that exists between an animal and a human being, perceived, developed and expressed through their mutual behaviour” [3]. A good human–animal relationship is not only positive from an ethical and moral point of view, bringing benefits to the animals in terms of well-being, but it is also extremely important for the farmer, since stressed animals have lower production levels and yield lower-quality products [4,5,6]. Farm animals may show fear of people by avoiding contact with them; they may be neutral, when the fear level is low, but still avoid contact; and they may have positive interactions, when fear is absent, when the animals allow physical contact [7]. The main aspects affecting human–animal relationships are genetic predisposition and husbandry practices that improve animals’ perception of humans, including habituation and early positive human contact [8].

Goats experiencing human contact from the first weeks of life are calmer and more docile and are more comfortable in the presence of humans compared to goats that have experienced minimal human contact; Boivin and Braastad (1996) [9] found that animals isolated one week after birth and subjected to positive treatment with humans spent a greater amount of time in contact with humans than isolated subjects not receiving the gentle treatment. Subsequently, non-stroked goats showed a lack of interest in humans and tended to avoid them, which negatively affected the well-being of the animals while also impairing the efficiency and manageability of procedures requiring direct contact with humans (milking, hoof trimming, etc.), and this was attributed to increased levels of fear [10]. Contact and relationships established with humans can have a strong impact, contributing to worsening or improving the animal’s living conditions. “Gentle” treatments carried out on pregnant goats promoted maternal care in the first hour postpartum, with increased frequency and duration of specific maternal behaviours such as grooming (cleaning the newborn from placental residues) and nosing (stroking the kid with the face) [11]. During their development, goat kids born from gentled goats displayed more play behaviours (jumping, often throwing random hind kicks, or throwing the head) and were faster in starting to suckle and to stand up on four legs [11]. In animals that were subjected to negative treatments, the onset of these behaviours can be delayed, suggesting a cognitive deficit that can have a negative impact on neonatal survival [11]. Traumatic events and/or stressful experiences can also have negative consequences for the foetus, especially in the phase in which specific organs and tissues (e.g., the brain) are formed. Such experiences have been found to increase cortisol levels in goats [11]. For instance, it has been observed that goats treated adequately by humans who have established emotional bonds with them within 20 days from the first approach have shown a significant increase in the size of the heart (a parameter directly related to weight gain) compared to a control group without such contact with humans [12]. Pregnant goats stressed by insufficient space allowance, as well as high stocking densities, can assume defensive or violent behaviours towards conspecifics and humans, and these behaviours can then be transmitted and acquired by their offspring, which will show greater fear than control goats when subjected to social and isolation tests [13]. Regarding the quality of the milk produced, in goats there is no evidence of the human–animal relationship affecting the final milk composition [12]. On the other hand, there may be effects regarding the percentage of milk not extracted from the udder following milking. In goats that had been raised by their mother, the amount of milk left in the udder at the end of milking was found to be greater than in those raised by humans, and which therefore had a relationship with them [14]; the milk yields were higher in the human-reared goats due to the lower quantity of residual milk left in the udder at the end of each milking. The reason given for this was that the goats raised by humans were accustomed to their presence from birth and therefore showed less fear of humans and a greater propensity to relate to them than the other goats.

A good human–animal relationship is fundamental to improving farm animal welfare, with associated health and production benefits in all farming systems, even those with improved precision and automation farming [8]. A human approach test is a welfare indicator included in practical on-farm protocols for welfare evaluation, such as the AWIN (Animal Welfare Indicators) project funded by Europe, for horses and sheep [15]; however, the AWIN protocol for goats relies on a stationary human test [16,17]. These protocols are useful for providing support to farmers by allowing them to quickly and efficiently assess the welfare of their animals using simple indicators.

In the approach tests, the aim is to evaluate the tendency of animals to approach a person familiar to them (or even an unfamiliar person). In the reaction tests to humans in motion, the avoidance distance is measured (in the pen or to the manger) when the detector walks at a constant pace towards an animal. In the reaction tests for farming practices, however, the behaviour of the animals during milking is directly observed, as this is a direct consequence of their degree of agitation and fear. In carrying out these tests the interference of environmental variables must always be minimized by defining meticulously the evaluation protocol. The behaviour of goats in the milking parlour in response to interactions with humans is an important measure that allows us to assess the degree of agitation of the animal during this operation. In fact, low levels of agitation are associated with a good human–animal relationship [18].

The degree of restlessness of animals in the milking parlour, for example, is not always related only to a condition of stress induced by the quality of the human–animal relationship, but can also be due to the presence of adjacent subjects, the presence of insects, malfunctioning of the system, milking, etc. [19].

This work aimed to evaluate the effectiveness of methods for assessing and monitoring the welfare of lactating goats; in particular, attention was paid to methods regarding the assessment of the human–animal relationship and the relationships between the behaviour of operators during milking and the reactions of animals to humans. An HAR (Human–Animal Relationship) protocol for goats was developed, which included three tests: avoidance distance inside the pen, avoidance distance to the manger [18], approach test within 2 min [20] and an assessment of the behaviour of stockmen and goats during milking.

## 2. Materials and Methods

### 2.1. Farms, Animals, and Procedure

Fourteen dairy goat farms located in the province of Potenza (Southern Italy) were used with a range of 67–450 lactating animals in the flock (mean ± SD = 211 ± 131). Before performing the experiments, each farmer was asked to sign a consent form to allow researchers to observe the animals and perform the tests. Goats on all farms were of a local breed, Rossa Mediterranean, the most common in this area, as has been described by Zumbo and Di Rosa (2010), and which is a breed capable of adapting easily to new environment [21]. Only the farms with a feeding alley were selected, allowing us to test the animals for avoidance distance at the manger. Data about herd size, milking routine, farm management, housing and milking parlour were collected. As suggested by Cochran (1977) [22], the number of goats selected from each farm depended on the size of the flock, as shown in Table 1. Two observers were trained to standardize the data collection technique before the experiment, which took place from September-November 2019. They were trained in the identification of stockman and goat behaviours and test performing on a farm not involved in the experiment.

The human–animal relationship was assessed by performing three different tests: avoidance distance from the manger, avoidance distance inside the pen and approach test within 2 min. During the afternoon milking, the behaviours of the stockman and the animals were observed. To avoid errors during data collection during the test, the observers always wore the same clothes and avoided noise or other factors that could have interfered with the test, such as pens not being quiet, and without mobbing behaviour. The goats were milked twice a day, and during the experimental period, the behaviours of the operators and the animals during milking were observed in the afternoon. The milking parlours designed were traditional for goats, with elevated parlours and parallel or side-by-side parlours with front stall rapid exits. The milking parlour stalls had layouts ranging from 12 × 12 to 24 × 24 for each of the milking parlours on farms visited.

On all farms, the testing sequence was the same: avoidance distance at the manger/pen and approach test within 2 min (morning) and behavioural observations during afternoon milking. On all farms, stockman and goat behaviours in the milking parlour were recorded by the same observer, whereas the stockman’s behaviour when moving the animals from the waiting area to the milking parlour was always observed by the second observer. The latter also conducted the three tests performed in the barn.

### 2.2. Avoidance Distance at the Manger and inside the Pen

Five minutes after the feed was distributed, the observer started the test. Before starting the test, the observer was positioned at a distance of 2 m from the manger and waited until a goat became aware of their presence, to avoid a sudden fear response.

Animals were approached by the test person in a standardized way, i.e., directly from the front, starting by walking slowly (1 m/s with steps of about 60 cm) with the arm protruding forward slightly (inclined at about 45°) and the back of the hand facing the muzzle of the animal. During the test, the operator, without staring into the goat’s eyes, continued walking until the animal withdrew or allowed their muzzle to be touched. If the goat accepted the touch on the muzzle/nose, the experimenter tried to stroke the cheek of the animal for at least 1 s but for not longer than 3 s. The avoidance distance was estimated at the moment of goat withdrawal as the distance between the observer’s hand and the animal’s head with a resolution of 10 cm. If the animal allowed the test person to approach within 10 cm without being touched, a distance of 0.1 m was recorded, while if the operator managed to touch the animal, the distance recorded was 0 m.

The test was carried out in succession on the animals that were at the feeding trough, but the subjects adjacent to those tested who reacted to the approach of the operator were not tested.

The following variables were calculated for each farm:-animals that allowed themselves to be approached until they were touched (avoidance distance = 0), %.-mean, median and range of the avoidance distance, m.

The evaluation of the avoidance distance inside the pen was used following the same procedure as the first test. In this case, the operator needed to enter the pen without generating a runaway reaction in the group, select an animal that was not in decubitus or was taking food or water and stand in front of it at a distance of 2 m. The operator made sure that the animal to be tested was aware of their presence to avoid sudden alarm reactions. The test was conducted in the same way as the previous one. The variables recorded for each farm were the same as those for the avoidance distance at the manger.

### 2.3. Approach Test within 2 Min inside the Pen

Subsequently, having completed the two previous tests, the operator conducted the approach test. The approach test consisted of the operator, before entering and walking inside the pen, waiting at the gate for 30 s, to avoid sudden alarm reactions of fear, and without staring into the goats’ eyes. The operator entered the pen close to the wall, and once the animals became calm again, walked slowly inside the pen (at 2 m/s with step lengths of about 60 cm), trying to touch as many animals as possible. The approach test within 2 min has been chosen for the speed and low technical challenge, during which period the observer recorded all animals that allowed themselves to be touched.

### 2.4. Stockmen and Goats’ Behaviour during Milking

The test began when the stockman started to move the animals to the parlour and the animals were observed until they left the parlour. The interactions, reported in Table 2, have been classified according to those provided by Waiblinger et al. (2002) [20] and Napolitano et al. (2019) [23] for the assessment of cattle and buffalo, respectively, and also reported were the use of sticks to encourage the movement or increase the speed of goats.

The goat behaviours, recorded from the entrance in the milking parlour to the removal of the milking cluster, were stepping (foot lifted less than 15 cm from the ground) and kicking (raised above 15 cm off from the ground, even if a clear kick was not visible). They were recorded whenever the stockman was within 0.5 m of the animals. The following interaction variables were recorded for each farm: positive interactions/goat, n; neutral interactions/goat, n; and negative interactions/goat, n; and positive interactions/total interactions,%, neutral interactions/total interactions,%, negative interactions/total interactions,%, as indicated by Waiblinger et al. (2002) [20] and Napolitano et al. (2019) [23], and the number of steps and kicks were recorded.

### 2.5. Statistical Analysis 

Data were analysed using the SAS software program (1990) [24]. The farm was used as the experimental unit. Descriptive statistics were used to generate the means, standard deviations, medians and ranges of all variables and the number of animals recorded. Following that, the Spearman correlation coefficient (*r_s_*) was used to estimate correlations; in this case, the median was used for two tests of avoidance distance at the manger and inside the pen.

## 3. Results and Discussion

### 3.1. Assessments of Goat Behaviours during Two Tests of Avoidance Distance and the Approach Test

Animals’ reactions in the presence of humans have always been useful to understand how they perceive people. The assessment of avoidance distance through tests for livestock animals such as cattle, buffalo and sheep [23,25,26,27] has provided a functional index for evaluating and measuring the quality of human–animal interactions and the fear of animals in the presence of humans [28].

The human–animal relationship in livestock animals is important not only from an ethical standpoint but also because it has allowed us to evaluate and classify farms and stockmen as “good” or “poor” in terms of animal welfare and farm management quality [29,30]. Jackson and Hackett (2007) [12] observed that goats were more easily approached by unknown people if they had previously had a positive experience with humans, compared to goats that had never experienced human interactions before.

Specific behavioural tests, such as avoidance distance at the manger and inside the pen, were used to assess fear and the relationships with human–goat interactions. A positive human–animal relationship is associated with a reduced distance in approach tests and a higher number of animals that allow themselves to be touched. Table 3 shows the mean values, the medians and the ranges concerning the avoidance distance at the manger and pen and the behaviours of the goats during milking.

Mersmann et al. (2016) [31] in 43 Austrian and German goat farms reported results similar to those observed in both buffaloes [23] and sheep [27]. The percentage of goats that let themselves be touched at the manger (mean 12.36 ± 9.50 SD) was higher than that found in the pen (mean 9.67 ± 11.86 SD) and was less variable; this could have been due to the presence of the feed inducing the animals not to move away from the manger. In addition, animals are more habituated to the presence of humans in the feeding alley than in the pen [27]. The percentage of goats that let themselves be touched during the approach test within 2 min inside the pen was similar to the percentage of the animals touched during a test of avoidance distance inside the pen (mean 8.19 ± 13.78 SD). Furthermore, it should be considered that the goat is a gregarious animal and, inside the pen, this gregariousness can greatly influence its behaviour, so the goats’ reactions to the approach of a human may in part be influenced by the behaviour of the other animals in the flock [32]. The goats’ avoidance distances (excluding those that allowed their muzzles to be touched) were 1.18 m (median) in the manger and 0.62 (median) meters in the pen. A study by Mattiello et al., (2010) [28] on 17 goat farms located in the province of Sondrio reported this distance to be 0.29 m, slightly higher than that observed by Mersmann et al. (2016), 0.26 m [31] in Austrian and German goat farms. In sheep, similar results to our study were reported by Napolitano et al. (2011) [27] on 20 farms in Basilicata. In a comparative study, the avoidance distances were found to be lower for goats than for cattle while conversely, the percentage of animals touched was higher in goats than in cattle [28]. This is probably due to the fact that goats, like buffaloes [23,33], are instinctively curious animals. Mattiello et al. (2010) [28] also showed that the avoidance distance is strongly influenced by farm size, with larger distances observed on larger farms. This may be due to the lack of mechanization on small farms, which results in goats more frequently coming into contact with stockmen during much of the farm’s management and husbandry operations.

### 3.2. Goats and Stockmen’s Behaviour during Milking

The number of steps and kicks made by the goats during milking was used to assess the animals’ level of restlessness. The frequency with which the animals raise their legs during milking is considered a measure of agitation, whereas the tendency to kick is a sign of discomfort, which may cause injuries. The number of steps in this study was greater than the number of kicks (Table 4), as was also reported for buffaloes by Napolitano et al. (2019) [23]. Furthermore, the number of steps and kicks was similar to what has been observed in cattle [20,34] and buffalo [23,35]. The frequency of steps and kicks could be influenced by various factors, including pushes from adjacent animals, hoof problems, mineral deficiencies, the presence of haematophagous insects and malfunctioning of the milking machine [1]. However, it is widely assumed that the state of agitation may be due to the presence of humans [35]. A strong correlation between the stockmen’s behaviour and the level of agitation in animals has been found in several studies [35,36,37].

In this study, 60% of the interactions were neutral, while the positive and negative interactions both had similar values, 20% (Table 4). Muri et al. (2013) [38] reported that goats showed no fear in the presence of an unknown human due to the positive interactions they had previously had with other humans.

In a study carried out on dairy cows, Rushen et al. (1999) [39] showed that the presence of a stockman with a negative attitude in the milking parlour caused greater agitation among the cows, particularly in subjects who could discriminate between and recognize single employees, whereas in subjects who were unable to do so, restlessness occurred with stockmen with both positive and negative attitudes. Therefore, it is essential that during milking, such positive interactions as talking in a low voice and caressing the animals with gentle touches and without making noises are all actions that can help to calm the animals, while screaming or hitting the animals forcefully may cause an increase in agitation and stress. This also has implications for production in dairy animals, as a high level of agitation in the milking parlour can increase cortisol and catecholamine secretions, resulting in an increase in the amount of residual milk in the udder [34]. It should be emphasized that in the scientific literature consulted on goats, there are no studies that have evaluated the human–animal relationship during milking operations Celozzi et al. [1], even if is known that it can have a negative effect on milk production [12,38].

### 3.3. Correlations of Variables Observed in Goats and Stockmen Behaviour

Regarding the degree of association between the behaviour of the milker and the animals during the milking routine, it was found that the percentage of neutral interactions tended to be positively correlated only with the number of kicks performed by the animals (Spearman correlation test (*r_s_*) = 0.476; *p* < 0.10), as has been shown in cattle [39], and with the percentage of goats moving when approached at a distance >1 m at the manger (*r_s_* = 0.799; *p* < 0.0006). The percentage of neutral milker interactions was negatively correlated with the number of animals permitting themselves to be touched at the manger (*r_s_* = −0.606; *p* < 0.04) and in the pen (*r_s_* = −0.612; *p* = 0.01). A negative correlation (*r_s_* = −0.669; *p* < 0.01) was found between the approach test and the number of goats that allowed themselves to be touched inside the pen within 2 min.

The avoidance distance at the manger was positively correlated with the number of kicks performed by the goats during the milking operation (*r_s_* = 0.539; *p* < 0.04) [40,41]. Finally, avoidance distance at the manger was positively correlated with avoidance distance in the pen (*r_s_* = 0.607; *p* < 0.01), as were the percentages of goats touched at the manger and in the pen (*r_s_* = 0.647; *p* < 0.01). No significant correlations were found for any of the other variables. This result suggests that goats, as with other farm animals [23,27], are able to generalize their response to humans, and if they perceive negative stimuli from the stockmen in the milking parlour, they also increase their avoidance response to an unknown person approaching them in the pen. In dairy cattle, it has been found that the positive attitude of milkers is positively correlated with the number of positive interactions [20] and negatively correlated with the number of negative interactions [18]. These results would indicate that even in the case of dairy goats, the attitude of stockmen during the milking routine affects the behaviour adopted towards the animals [36].

In the case of this study, it should be noted that the definition of neutral interactions used has a negative connotation. In fact, our definitions of positive, neutral and negative interactions correspond to what suggested by Breuer et al. (2000) [34] and Hemsworth et al. (2000) [18] for cattle for positive, negative and very negative interactions. Similar results were observed in cattle [18,34] and buffalos [22]. Furthermore, in a study carried out on sheep, it was observed that the avoidance distance to the manger was negatively correlated with the adoption of negative behaviour by milkers [26].

### 3.4. Limitations of the Study

The small number of farms (*n* = 14) where the study was performed could be a limitation to the findings of this study. All goats in this study were polled, but horned animals might respond differently.

## 4. Conclusions

The results obtained from tests of avoidance distance at the manger and pen suggest that both methods are valid to evaluate the human–animal relationship in goats, but that where the feed alley exists, it is preferable to use avoidance distance at the manger because it is less expensive in terms of time spent on the evaluation. Furthermore, as demonstrated in other animal species, it is confirmed that stockman behaviour may affect goat behaviour.

## Figures and Tables

**Table 1 animals-13-02030-t001:** Descriptive statistics (mean ± standard deviation, range) of animal numbers observed during the tests and operators of the milking parlour (*n* = 14).

Variables	Mean	Min–Max
Flock size: lactating animals, *n*	211 ± 131	67–450
Lactating pen, *n*	2.43 ± 1.3	1–5
Animals observed at milking parlour/farm, *n*	93 ± 58	30–200
Milkers observed/farm, *n*	1.7± 0.6	1–3

**Table 2 animals-13-02030-t002:** Classification of voice and tactile interactions used by the milker [20,23].

Interactions	Positive	Neutral	Negative
Sounds	Speaking calmly, with a soft voice and long-lasting sounds	Speaking in an authoritative tone: decisive tone with short sounds, used to incite or stop movement	Speaking impatiently, harshly and quickly with short sounds
		Clapping hands, whistling, hitting structures with the stick	Screaming very loudly and harshly; used to incite or stop movement
Tactile	Patting	Hand contact and little use of force, possible low noise used to facilitate movement	Hitting animals with the hand with moderate to high force, producing distinct sounds
	Gentle touches, hand resting on the animal without making noise	Light strokes with the stick, little use of force with no or very low sounds	Striking with a stick with moderate to high force, striking with kicks

**Table 3 animals-13-02030-t003:** Means (±standard deviation), medians and ranges of variables observed of animals during the tests (*n* = 14).

Variables	Mean	Median	Min–Max
Avoidance distance at the manger, m	0.63 ± 0.28	0.62	0.17–1.25
Avoidance distance inside the pen, m	1.19 ± 0.58	1.18	0.37–2.00
Animals that can be touched at the manger, % (0 m)	12.36 ± 9.50	12.35	0.00–31.81
Animals that can be touched in the pen, % (0 m)	9.67 ± 11.86	1.43	0.00–32.00
Animals that can be touched within 2 min inside the pen	8.19 ± 13.78	1.43	0.00–50.00

**Table 4 animals-13-02030-t004:** Means (±standard deviation), medians and ranges of the variables observed in the stockmen and goats in the milking parlour.

Variables	Mean	Median	Min–Max
Positive interactions/goat, *n*	0.05 ± 0.07	0.024	0.00–0.23
Neutral interactions/goat, *n*	0.29 ± 0.31	0.14	0.01–0.97
Negative interactions/goat, *n*	0.12 ± 0.19	0.04	0.00–0.66
Positive interactions, %	19.64 ± 28.98	5.36	0.00–83.30
Neutral interactions, %	59.76 ± 25.45	56.25	16.67–100.00
Negative interactions, %	20.60 ± 23.32	13.64	0.00–59.25
number of steps/goat, *n*	1.12 ± 1.22	0.86	0.11–5.03
number of kicks/goat, *n*	0.73 ± 1.06	0.34	0.03–4.08

## Data Availability

The raw data have not been published or stored elsewhere but are available on request from F.C.

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
