# Peer review of "Human–Animal Interactions in Dairy Goats"

_animals, 2023, doi:10.3390/ani13122030_

Round 1

Reviewer 1 Report

Comments attached

Author Response

Reviewer 1 and Reply to Reviewer 1

Thank you very much for your comment and suggestions, we appreciated.

Below are described changes made.

A brief summary

The study is a strait forward field investigation of the possible use of four methods of evaluating dairy goat welfare that are not currently enshrined in the AWIN welfare assessment protocol (1). The study examines 3 human movement (vs human stationary) animal welfare (HAB) indicators, none of which are included in the AWIN-AP. The three tests were two human movement INDIVIDUAL ANIMAL tests (animal avoidance manger and animal avoidance pen); one human movement GROUP ANIMAL test (the human touch test 180s) where the reference AWIN test is human stationary approach test 300s. The final test was a human husbandry behavioural – lactating goat evaluation at milking which is not easily categorized under the AWIN template (2). Results suggest avoidance tests correlate with each other, rough handling results in standoffish goats and novel discovery that animal avoidance manger was as reliable as animal avoidance pen, which probably is of value to future on farm welfare assessments.

General concept comments

Animal avoidance tests are forefront of this study and the authors fail to mention that pens of goats must be of a quiet but not mobbing nature to complete an in pen avoidance test (3) and the finding that the manger variant of the test is as reliable is good news.

Thank you for your suggestions, in the L166 is described “To avoid errors during the data collection during the test, the observers always wore the same clothes and avoided noise or other factors that could have interfered with the test”. We added a sentence about the quietness of pens where were made the tests. In L183 “Before starting the test, the observer was positioned at a distance of 2 m from the manger and waited until a goat became aware of their presence, to avoid a sudden fear response”

In Table 3, avoidance at the manger is less variable than avoidance in the pen which should be pointed out in the discussion.

Thank you we added this in L275 as suggested.

The breed of goat is different from most of the studies done previously. Some mention of temperament of this breed would be useful if there is a supporting reference.

Thank you for this comment, “the breed of Rossa mediterranea is common in this area in Italy. This is a goat characterised by its robustness and ability to adapt readily to new environments and in addition, it has a high milk yield and is a prolific animal.” We added in a sentence L148.

Zumbo, Alessandro & Di Rosa, Ambra. (2010). Effects of parity and type of kidding on the quantitative and qualitative milk characteristics of “Rossa Mediterranea” goats. Italian Journal of Animal Science. 6. 10.4081/ijas.2007.1s.636.

In this type of fieldwork, I would expect a couple of paragraphs on limitations of the study. Are farms with a feeding alley significantly different from farms with other feeding infrastructure? In this case, 14 farms is not a powerful sample size, there is no subsample to evaluate the repeatability of measurements and there is no possibility of drawing causal relationships between measured indices. Things like all goats were polled or the proportion of horned goats in a pen, participation in health improvement programs for mastitis, paratuberculosis, individual animal performance records or CAE would give the reader a flavour of the farming culture and production system.

Thank you, limitations of the study have been added in L369.

Details of the methodology are deficient.

  1. There is no reference to how many animals per pen (avoidance pen test) and how many pens per farm were selected. The AWIN standard has a cohort size based sampling frame for individual tests in pen setting see Table 3 in (1). And there may remain a never lower than 40% of the pen examined in individual assessment (welfare Quality 2009) assumption (4).

Thank you for that, we specified that in Table 1.

  1. The forced human approach test 120s, is a unusual choice for test as it is poorly documented. The article reference 10 (Waiblinger, S et al 2006 AABS) is a review with limited description or reference to this test. I could not find a reference to it’s use in goats (some pig application). The paper need a reason for choosing this test I presume because and speed and low technical challenge.

Further information has been added L215.

  1. I did not notice in the report the number of pens/farms nor any attempt to control for or evaluate the variation between pens. The total of animals subjected to individual tests is irrelevant if a single value mean+sd is what is reported in Table 3. This would be clear by inserting an n=14 somewhere in the title or in the table.

Thank you. We added the number to the title of Table 3.

Specific comments

Line 354 These results would indicate that even in the case of dairy goats, the attitude of stockmen during the milking routine affects the behaviour adopted towards the animals [33].

 How was the attitude of stockmen evaluated independently of their behaviour during the push-up and milking? This statement may be inaccurate or a bit of an overreach for the data in this paper or circular reasonoing.

Thank you for your comment This is from a different study and is part of our discussion putting our results into context. Line 354 is: A negative correlation (rs = - 0.669; p<0.01) was found between the approach test and the number of goats that allowed themselves to be touched inside the pen within 2 minutes. Stockmen behavior is always difficult to evaluate, the descriptors we used are we think clear enough for readers to understand.

The manuscript is clear and well written and appropriate to the subject.

 The citations are older but, this reflects the reality of the research in the field. In lines 68-79 there is a cluster of statements related to intrauterine stress affecting offspring. This was all cited to a single paper which I have never seen reproduced and if it has been reproduced, the more recent confirmatory paper should be cited.

The paper referenced can be found here: https://pubmed.ncbi.nlm.nih.gov/26850289/ We don’t think that 2016 is so long ago.

I am not certain that the introduction is efficient as referencing studies measuring cortisol and the like are of questionable utility to a paper where those variables are not measured.

Yes, true we didn’t measure cortisol but giving some background to welfare evaluation without mentioning cortisol would be deficient.

This type of work seldom tests a hypothesis and looks to more practical application and feasibility so for this subject, I believe the methodology is sound and repeatable in other production geographies. I could identify no ethical concerns with this project although there was no description of how individual farms were included in the study and how permission to use results was obtained. It is possible to have ethical concerns outside of those of animal use.

Thank you about it, in L 148 we described why we choose those farms, we added a sentence about permission in the methodology section in L145 “Before performing the experiments has been asked to each farmer to sign a consent form, to allow to observe the animals and perform the tests.”

Reviewer 2 Report

It is a very interesting work on human and animal interaction in dairy goats. It is well supported by existing knowledge on the subject and has been carried out based on a large sample. The article is well organized and well written and, as such, is simple to follow. However, some corrections are necessary to improve its clarity. Accurate references support the text. The Tables are essential for understanding the article. The material and methods are clearly described, which allows a perfect understanding of what has been done. The results are well presented and well discussed. Finally, the results corroborate the conclusions.

Some detailed comments are below:

L67-68 human-animal relationship are genetic predisposition, “change with” human-animal relationships are genetic predisposition

L163 milking parlour designed were traditional “change with” milking parlours designed were traditional

L174 after feed was distributed “change with” after the feed was distributed

L247 animals such as “change with” animals, such as

L273 habituated at the presence “change with” habituated to the presence
L285 comparative study the “change with” comparative study, the

L301 Napolitano et al “change with” Napolitano et al.

L305 of haematophagic insects “change with” of haematophagous insects

L306 may be “change with” maybe

L342 Celozzi et.al. [1], even if is known that “change with” Celozzi et al. [1], even if it is known that

L370 to what were suggested “change with” to what was suggested

L372 observed in cattle) “change with” observed in cattle

L417 J. Dairy Sci “change with” J. Dairy Sci.,

L429 Physiology & Behavior, 2016, 157: 129-138 “change with” Physiol. Behav. 2016, 157, 129–138

L450 Animals (Basel). 2019 Jun 28;9(7):396. “change with” Animals 2019, 9, 396

Please check all reference section

English is ok

Author Response

Reviewer 2 and Reply to Reviewer 2

It is a very interesting work on human and animal interaction in dairy goats. It is well supported by existing knowledge on the subject and has been carried out based on a large sample. The article is well organized and well written and, as such, is simple to follow. However, some corrections are necessary to improve its clarity. Accurate references support the text. The Tables are essential for understanding the article. The material and methods are clearly described, which allows a perfect understanding of what has been done. The results are well presented and well discussed. Finally, the results corroborate the conclusions.

Thank you very much for your comment and suggestions, we appreciated. We changed all as suggested.

L67-68 human-animal relationship are genetic predisposition, “change with” human-animal relationships are genetic predisposition

Yes, agreed

L163 milking parlour designed were traditional “change with” milking parlours designed were traditional

Yes, agreed

L174 after feed was distributed “change with” after the feed was distributed

Yes, agreed

L247 animals such as “change with” animals, such as

Actually no, no comma needed here

L273 habituated at the presence “change with” habituated to the presence

Yes, agreed

L285 comparative study the “change with” comparative study, the

Actually no, no comma needed here

L301 Napolitano et al “change with” Napolitano et al.

Yes, agreed

L305 of haematophagic insects “change with” of haematophagous insects

Yes, agreed

L306 may be “change with” maybe

 No, may be is intended.

L342 Celozzi et.al. [1], even if is known that “change with” Celozzi et al. [1], even if it is known that

Yes, agreed

L370 to what were suggested “change with” to what was suggested

No, “definitions” are plural and this is what “were” agrees with.

L372 observed in cattle) “change with” observed in cattle

Yes, agreed

L417 J. Dairy Sci “change with” J. Dairy Sci.,

Yes, agreed

L429 Physiology & Behavior, 2016, 157: 129-138 “change with” Physiol. Behav. 2016, 157, 129–138

Yes, agreed

L450 Animals (Basel). 2019 Jun 28;9(7):396. “change with” Animals 2019, 9, 396

 Yes, agreed

Reviewer 3 Report

This work aims to evaluate the effectiveness of methods to assessing and monitoring the welfare of lactating goats, in particular regarding the human-animal relationship. The strengths of the work lie in having rigorously applied the AWIN protocols to this species to understand the level of welfare in the farm environment. The main weakness lies in the number of data used, in particular the correlations are calculated on 14 data and therefore the significance found is questionable. One can choose whether to leave the calculation at the simple descriptive level (without significance) or to insert a sentence highlighting that the reported results are not particularly robust. Otherwise the work is well written and has a good scientific sound.

Minor specific comments are below:

Line 27: specify in the abstract the unit of measurement of the avoidance distance

Line 125: the first time the acronym is used, it must be specified

Line 141: in the table identifier in the brackets, it is also necessary to insert range (mean ± standard deviation, range)

Line 318: first part of the sentence is missing

Author Response

Reviewer 3 and Reply to Reviewer 3

Thank you very much for your comment and suggestions, we appreciated them. We changed all as suggested.

This work aims to evaluate the effectiveness of methods for assessing and monitoring the welfare of lactating goats, in particular regarding the human-animal relationship. The strengths of the work lie in having rigorously applied the AWIN protocols to this species to understand the level of welfare in the farm environment. The main weakness lies in the number of data used, in particular, the correlations are calculated on 14 data and therefore the significance found is questionable. One can choose whether to leave the calculation at the simple descriptive level (without significance) or to insert a sentence highlighting that the reported results are not particularly robust. Otherwise, the work is well written and has a good scientific sound.

We add a sentence recognizing the small number of samples, at the end of the results section

Minor specific comments are below:

Line 27: specify in the abstract the unit of measurement of the avoidance distance

This added, m

Line 125: the first time the acronym is used, it must be specified

HAR protocol. Now added HAR (Human-Animal Relationship

Line 141: in the table identifier in the brackets, it is also necessary to insert range (mean ± standard deviation, range)

Agreed, now reads: Descriptive statistics (mean ± standard deviation and range) of animal numbers observed

Line 318: first part of the sentence is missing

Yes, thanks for noticing this, should now read: In a study carried out on dairy cows.